# Synthesis, Characterization and Non-Isothermal Crystallization Kinetics of a New Family of Poly (Ether-Block-Amide)s Based on Nylon 10T/10I

**DOI:** 10.3390/polym13010072

**Published:** 2020-12-27

**Authors:** Xin Tong, Zhao Wang, Mei-Ling Zhang, Xiao-Jun Wang, Gang Zhang, Sheng-Ru Long, Jie Yang

**Affiliations:** 1College of Polymer Science and Engineering, Sichuan University, Chengdu 610064, China; tongxin18328582234@163.com (X.T.); wangzhao2019@outlook.com (Z.W.); 2Analytical and Testing Center, Sichuan University, Chengdu 610064, China; zhangmeilincn@126.com (M.-L.Z.); gangzhang@scu.edu.cn (G.Z.); lsrhome@163.com (S.-R.L.); 3State Key Laboratory of Organic–Inorganic Composites, Beijing University of Chemical Technology, Beijing 100029, China; 4State Key Laboratory of Polymer Materials Engineering, Sichuan University, Chengdu 610065, China

**Keywords:** thermoplastic elastomer, non-isothermal crystallization kinetics, semi-aromatic polyamide

## Abstract

A series of novel thermoplastic elastomers based on (poly(decamethylene terephthalamide/decamethylene isophthalamide), PA10T/10I) and poly(ethylene glycol) (PEG) were synthesized via a facile one-pot, efficient and pollution-free method. The thermal analysis demonstrates that the melting points of the resultant elastomers were in the range of 217.1–233.9 °C, and their initial decomposition temperatures were in the range of 385.3–387.5 °C. That is higher than most commercial polyamide-based thermoplastic elastomers. The tensile strength of the resultant elastomers ranges from 21.9 to 41.1 MPa. According to the high-temperature bending test results, the resultant samples still maintain considerably better mechanical properties than commercial products such as Pebax^®^ 5533 (Arkema, Paris, France), and these novel thermoplastic elastomers could potentially be applied in high-temperature scenes. The non-isothermal crystallization kinetics of the resultant elastomers and PA10T/10I was investigated by means of Jeziorny and Mo’s methods. Both of them could successfully describe the crystallization behavior of the resultant elastomers. Additionally, the activation energy of non-isothermal crystallization was calculated by the Kissinger method and the Friedman equation. The results indicate that the crystallization rates follow the order of TPAE-2000 > TPAE-1500 > PA10T/10I > TPAE-1000. From the crystallization analysis, the crystallization kinetics and activation energies are deeply affected by the molecular weight of hard segment.

## 1. Introduction

Thermoplastic polyamide elastomers (TPAE), as a new variety of thermoplastic elastomers (TPEs), have received extensive attention with excellent performance and wide commercial applications [1,2,3,4]. Generally, a TPAE is a multiblock copolymer composed of statistically alternating soft and hard segments. The soft and hard segments tend to be microphase-separated from each other with the difference of hydrophobicity and hydrophilicity [3]. The soft segment, acting with high elasticity to TPAE, presents a random coil configuration [1,4,5]. The hard segments, acting as physical cross-linking points, are either glassy or crystalline at room temperature [6,7]. At present, aliphatic polyamides such as PA-6, PA-11 and PA-12 are selected as the hard segments for commercial TPAEs [5,8,9]. Compared with other hard segments, polyamides possess much higher hydrogen bond density and stronger intermolecular forces, endowing the elastomer with a high melting temperature (Tm), good corrosion, thermal and abrasive resistance, etc. [10].

In recent years, aliphatic polyamides have been widely applied in various fields because of their high tensile strength and excellent chemical resistance [11,12,13]. However, the weaknesses of aliphatic polyamides become obvious, such as high moisture absorption and poor thermal properties [14,15]. To improve the heat resistance and strength of aliphatic polyamides, aromatic rings are incorporated into the backbones of polymers. According to this strategy, many kinds of semi-aromatic polyamides have been developed and become commercially available, such as poly(hexamethylene-terephthalamide) (PA6T) [16,17], poly(nonamethylene-terephthalamide) (PA9T) [18,19] and poly(dodecamethylene-terephthalamide) (PA12T) [20]. Especially PA9T is found to have good comprehensive performance. However, it is limited by the high cost and low production. Poly(decamethylene terephthalamide) (PA10T) is synthesized from 1, 10-decanediamine and terephthalic acid [18], having a similar performance as PA9T. Moreover, the diamines of PA10T can be prepared from castor oil, which has a relatively lower cost than PA9T [21,22]. Combined with the excellent comprehensive performance of PA10T, it can be potentially incorporated into the molecular chain of TPAEs as the hard segment [23,24]. However, there are few literature reports about TPAEs based on semi-aromatic polyamides, especially for PA10T and its copolymer.

To date, the synthesis methods of TPAEs have been widely studied and two mainstream methods are used to synthesize TPAE [25]. One is the diisocyanate method where diisocyanate-terminated polyamide and a soft segment, such as poly(ethylene glycol) (PEG), are added into a solvent and react at a low temperature. The core issue of this method is that diisocyanate reacts from the end group serving as a bridge between the hard segment and soft segment [26]. According to this method, many kinds of TPAE have been successfully fabricated [5,10]. The advantage of this method is that the condition of reaction is mild [6]. However, the reaction is accompanied by the production of CO_2_, which is very difficult to remove with the high viscosity of the copolymer [10]. Moreover, the solvent is difficult to recycle. This method is generally not used for commercial TPAEs. The other one is the melt polycondensation method based on polyamide with binary carboxyl end-groups and polyether [27]. In general, this method is divided into two steps. The first step is to synthesize carboxyl-terminated polyamide hard segment with a proper molecular weight. The next step is polycondensation with soft segment under high temperature and high vacuum conditions. Melt polycondensation has the advantages of high efficiency, low cost and little pollution, so most commercial TPAEs use this one [9]. In this research, a more facile and efficient method, a one-pot route, is explored to prepare PA10T-based TPAE. This method has only one feeding process whereby all the ingredients are added into a reactor at the same time. Compared with the traditional two-step melt polycondensation method, the one-pot method requires less production equipment, it could save the time of switching between devices and could improve the production efficiency.

Crystallization is a critical factor which determines physical, chemical and mechanical properties of semi-crystalline block polymers [28]. The crystallization process is affected by several key factors, one of which is crystallization kinetics. Therefore, doing further research on the crystallization behavior can be of service to analyze crystalline structures, degree of crystallinity and processing conditions of polymeric materials [29]. Generally, there are two kinds of crystallization processes in polymers: isothermal crystallization [30] and non-isothermal crystallization [31,32]. However, industrial process such as extrusion and injection molding are carried out under dynamic and non-isothermal crystallization conditions [33], and there has been little discussion about non-isothermal crystallization kinetics of TPAE, especially for semi-aromatic polyamide-based TPAEs.

In this work, a series of TPAEs based on PA10T/10I (poly (decamethylene Terephthalamide/decamethylene isophthalamide)) were firstly prepared by a facile and efficient one-pot synthetic method. The copolymerization ratio of PA10T/10I 60/40 was selected for as the hard segments, according to our previous research [34]. The properties of PA10T/10I-based TPAEs were comprehensively investigated via FT-IR, ^1^H-NMR, differential scanning calorimetry (DSC), thermogravimetry analysis (TGA), tensile test and high temperature flexural test. The relationship between the thermal properties of TPAEs and the hard segment molecular was deeply explored, and the mechanical properties at high temperatures were studied contrastively. Additionally, the non-isothermal crystallization behaviors of resultant TPAEs were firstly studied in detail. Jeziorny [31] and Mo’s methods [35] were used to analyze the influence of cooling rate on crystallization behaviors. Furthermore, the activation energies of non-isothermal crystallization of the resultant polymers were calculated by Kissinger [36] and Friedman methods [28,35].

## 2. Experimental

### 2.1. Materials

The following materials were used to synthesize PA10T/10I-based TPAEs. Terephthalic acid (TPA), isophthalic acid (IPA) and 1,10-Diaminodecane (DMD) were supplied by Yangzi Petrochemical Co., Ltd. (Nanjing, China), Lotte Chemical Co., Ltd. (Seoul, Korea) and Wuxi Yinda Nylon Co., Ltd.(Wuxi, China), respectively. Deuterated trifluoroacetic acid (TFA-D), Polyethylene glycol (PEG, Mw = 1000g/mol) and tetraisopropyl orthotitanate (Ti (i-OC4H9)4) were obtained from Aladdin Reagent Co., Ltd. (Shanghai, China). Sulfuric acid was purchased from Kelong Chemical Reagent Co., Ltd. (Chengdu, China). Pebax^®^ 5533 was obtained from Arkema Inc (Paris, France). All reagents were commercial grade and used as received without further purification.

### 2.2. Synthesis of Thermoplastic Elastomers Based on PA10T/10I and PEG

The synthetic route of TPAEs is presented in Scheme 1. The molecular weights of the hard segment PA10T/10I were selected as 1000, 1500 and 2000 and marked as TPAE-1000, TPAE-1500 and TPAE-2000, respectively. The feeding ratio of hard and soft segment, and the feeding mass of ingredients are listed in Table 1. All reactions were carried out in a 2-L stainless steel reactor with mechanical stirring and a nitrogen inlet and outlet. Take TPAE-1000 as a typical example—TPA (355.73 g), IPA (273.15 g), DMD (451.00 g) and PEG (949.49 g) were added to the autoclave at one time. After piping nitrogen gas for 5 min, the mixture was stirred at 265 °C and held for 2 h. Then, the pressure was decreased to 100 Pa to remove oligomer and excessive water, and the temperature of the reaction system was further heated up to 275 °C for another 3 h. Finally, the resulting TPAE polymers were dried in a vacuum oven at 80 °C for the next experiment. PA10T/10I was synthesized in the same way, and the ratio of 10T and 10I was selected as 60:40 for better comparison.

### 2.3. Characterization

#### 2.3.1. FT-IR and ^1^H-NMR

To verify the structure of the TPAEs, FT-IR and ^1^H-NMR experiments were conducted. For the FT-IR experiment, a piece of square film about 1 cm^−1^ on a side was prepared to collect the FT-IR spectra using a Nicolet 670 spectrophotometer (Thermo Fisher, Waltham, MA, USA) in reflection mode. The scanning range was from 4000 to 500 cm^−1^. In ^1^H-NMR experiments, about 5-mg samples were dissolved in trifluoroacetic acid-D (CF_3_COOD), and ^1^H-NMR spectra were collected using a Bruker Avnance-600 NMR spectrometer (Karlsruhe, Germany) with tetramethyl silane (TMS) as the internal standard.

#### 2.3.2. Differential Scanning Calorimetry (DSC)

DSC measurements were conducted on Mettler Toledo DSC3 equipment (METTLER, Zurich, Switzerland). The process of the experiment was as follows: a dried sample of about 5 mg was first heated from 30 °C to 300 °C at a rate of 10 °C/min and kept for 5 min to eliminate thermal history. Then, the DSC data were subsequently collected at cooling and heating cycles at a rate of 10 °C/min from 0 °C to 300 °C, at a rate of 10 °C/min.

#### 2.3.3. Thermogravimetry Analysis (TGA)

TGA experiments were carried out on TA Q500 thermo balance from TA Co. (New Castle, DE, USA). Dried samples of about 5 mg were put into the aluminum pan and heated from room temperature to 800 °C at a rate of 10 °C/min in N_2_ atmosphere.

#### 2.3.4. Polarized Optical Microscopy (POM)

The crystalline morphology of the resultant samples was observed with the polarized optical microscope Olympus (BX61) (Aizu, Japan), which was equipped with Linkam hot stage (CSS 450). All the samples were firstly heated to 280 °C for 5 min to erase the thermal history, and then cooled to 100 °C at 5, 10 and 20 °C/min. The micrographs were collected at a certain interval.

#### 2.3.5. Mechanical Properties

Tensile measurements were performed on an electromechanical universal testing machine (MTS SYSTEMS CHINA CO., LTD., Shenzhen, China) and dumbbell-shaped specimens (25 × 4 × 2 mm) were prepared for testing. The tensile speed was 50 mm/min at room temperature according to ISO 527-1993. For the high-temperature bending test, TPAEs and Pebax^®^ 5533 with standard size (80 × 10 × 4 mm) were prepared by injection molding, and the tests were carried out on the same testing machine in bending mode. A temperature-controlled chamber was employed to control the test temperature which was in the range from 50 to 130 °C. The speed of cross-head was 2 mm/min and the span of the bending test was 32 mm according to ISO 178.

#### 2.3.6. Non-Isothermal Crystallization Kinetics

The non-isothermal crystallization of the TPAEs was performed on a Mettler Toledo DSC3. The testing control process was as follows: dried samples of about 5 mg were heated from 30 °C to 300 °C at a rate of 10 °C/min and held for 5 min to eliminate thermal history. Then, the temperature was cooled down to 30 °C at a specified cooling rate of 5, 10, 20 and 40 °C/min^−1^.

## 3. Results and Discussion

### 3.1. Characterization of TPAEs

FT-IR experiments were performed to detect the chemical structure of the TPAEs. Test results are presented in Figure 1; the absorption peaks at 3290 cm^−1^ are corresponding to the stretching vibration of N–H [15]. The peaks at 2920 and 2860 cm^−1^ are assigned to C–H stretching vibration. It can be found that the stretching vibration bands of C=O groups of the –COO– group appear correspondingly at 1729 cm^−1^ for all the samples, which confirms that the hard segments are connected through ester bonds [9]. The sharp peaks at 1622 and 1537 cm^−1^ are due to C=O stretching vibration (amide I) and CO–N–H bending vibration (amide II), respectively [28]. The absorption peaks at 1110 cm^−1^ indicate the characteristic absorption peaks of the ether bond C–O–C of PEG [37].

In Figure 2, ^1^H-NMR spectroscopy and the chemical structure of TPAEs and PA10T/10I are presented. Protons corresponding to H (a_1_) on the para-substituted benzene ring appear at 7.96–8.08 ppm. The aromatic protons (a_2_, a_3_ and a_4_) on the meta-substituted benzene ring appear in the regions 8.42–8.48, 8.11–8.17 and 7.75–7.82 ppm, respectively. The peaks at 3.70–3.82, 1.81–1.99 and 1.40–1.60 ppm are related to methylene groups CH_2_ (b), (c) and (d) in the DMD unit, respectively. Protons from methylene groups CH_2_ (e) which are assigned to PEG soft segments appear at 4.39–4.17 ppm. Protons from methylene groups CH_2_ (f) and CH_2_ (g) which connect with the ester bond generated from esterification reaction of PEG and hard segment appear at 4.19–4.27 and 4.75–4.82, respectively. Compared with TPAEs, PA10T/10I lack the chemical shift of (e), (f) and (g). From the results of FT-IR and ^1^H-NMR, it becomes apparent that PA10T/10I-PEG with different soft segment contents has been successfully synthesized.

### 3.2. Thermal Properties of TPAEs

Thermal property is a major factor in polymer processing. Figure 3 shows the second heating curves of TPAEs and PA10T/10I. The results are summarized in Table 2. It is easy to find that all curves have two melting peaks, implying that these four samples have double-melting behavior. This phenomenon exists commonly in semi-crystalline polymers and might be attributed recrystallization during the subsequent heating. On the other hand, the melting point of TPAEs is slightly lower than that of PA10T/10I. As the molecular weight of the hard segment decreases, the melting peak becomes broadened and shifts to the low-temperature region, following the order of TPAE-2000 (233.9 °C), TPAE-1500 (230.3 °C) then TPAE-1000 (229.8 °C). It may be explained that as the molecular weight of the hard segment decreases, the content of PEG segment relatively increases, which will disrupt molecular regularity and reduce the crystallization ability of TPAEs. Otherwise, the shorter hard segment molecule means a reduction in intermolecular forces and decreasing hydrogen bond density.

A TGA instrument was used to assess the thermal stability. The curves of four samples are shown in Figure 4a and the specific values are summarized in Table 2. It is easy to find that the decomposition temperature at 5% weight loss (T_d, 5%_) of PA10T/10I is 426.3 °C, while TPAEs show significantly lower values ranging from 385.3 to 387.5 °C. In addition, the decomposition rates of PA10T/10I and TPAEs are shown in Figure 4b. It is obvious that PA10T/10I exhibits a single weight-loss stage, while TPAEs have a double mass-loss stage, indicating a two-step degradation mechanism. The variation of the DTG (Differential thermal gravity) curve around 380–430 °C is due to the decomposition of PEG segment [5]. According to the previous literature, the degradation mechanism of aliphatic polyester segments is that ester linkage will break and decompose at the alkyl–oxygen bond, followed by pyrolysis around 370–440 °C [38]. Compared to the DTG curve of PA10T/10I, the variation of TPAEs curves near 450–480 °C. The main reflect factor is the hard segment PA10T/10I decomposition. Although the initial decomposition temperature of TPAEs is lower than that of PA10T/10I, the thermostability of PA10T/10I-PEG is better than most thermoplastic polyamide elastomer, such as PA6-PEG (320 °C) [5], PA6-PTMG (364 °C) [37], PA6-b-PDMS (339 °C) [10] and PA610-ester (344 °C) [39].

### 3.3. Crystal Structure of TPAEs

The crystal structure of TPAEs and PA10T/10I was observed using XRD, and the results are shown in Figure 5. In this figure, all the samples showed three strong reflections at 2θ = 19.7°, 21.0° and 22.3°, which correspond perfectly with the pattern of PA10T [21]. These results suggest that the hard segment molecular chain in TPAEs tends to form the crystal structure of PA10T. Otherwise, as the molecular weight of hard segment increases, the scattering peaks become sharper, indicative of the development of an ordered crystalline structure [9]. These results are in agreement with the growing trend of ΔH obtained from DSC measurements.

### 3.4. Mechanical Properties

As shown in Figure 6, the typical stress–strain curves are collected for PA10T/10I and TPAEs. The values of tensile strength and elongation at break are listed in Table 2. It can be seen from Figure 6 that PA10T/10I has an obvious yield point while TPAEs do not. It means that TPAEs have typical elastomeric behavior. Moreover, one can easily find that the tensile properties of TPAEs change as the soft segment content changes, which means that the mechanical properties can be tuned according to usage scenarios. As the soft segment content in TPAEs increases, the elongation at break increases, but the tensile strength decreases. In particular, the tensile strength and elongation at break of TAPE-1000 are 21.9 MPa and 898.8%, respectively. The reason for the low tensile strength of TPAE-1000 is that the intermolecular force decreases as the hard segment molecular weight decreases. Otherwise, a higher soft segment endows a better toughness of the material, resulting in higher elongation at break. In order to make a comparison, a summary of the key published results of polyamide-based thermoplastic elastomers can be found in Table 3.

In order to emphasize the advantage of TPAEs, Pebax^®^ 5533, which has similar mechanical properties as TPAE-1000 (Appendix A), was chosen to be a contrasting sample to further discuss the TPAEs’ properties. Figure 7 shows the flexural modulus of TPAEs and Pebax^®^ 5533 as a function of testing temperature (the specific test results are shown in Appendix A). In this figure, the flexural modulus of the four samples showed a similar trend with testing temperature changes. As the temperature increases, the flexural modulus of four samples decreases. When a specific temperature is reached, a turning point occurs where the value of the flexural modulus drops rapidly [34]. For Pebax^®^ 5533, the turning point is 70 °C, while the turning point of TPAEs occurs at 100 °C. It is worth noting that Pebax^®^ 5533 and TPAE-1000 have similar starting values; as the temperature increases, the flexural modulus of TPAE-1000 declines less than Pebax^®^ 5533, and the turning point of TPAE-1000 comes even later, meaning that TPAE-1000 has better retention of elastic properties than Pebax^®^ 5533 at high temperature conditions. This result corresponds to the previous work [34]; the reasons could be that the hard segment of TPAEs is composed of a semi-aromatic polyamide, which has higher heat resistance than aliphatic polyamide. These results indicate that the resultant TPAEs can replace Pebax^®^ in a high-temperature environment and have the potential to be widely used in various industrial fields.

### 3.5. Nonisothermal Crystallization Kinetics

Generally, non-isothermal crystallization kinetics has an important directive in production and application, which is beneficial to improve the performance of polymers. However, there are few studies that focus on polyamide-based thermoplastic elastomers, and many less in semi-aromatic polyamide-based TPAEs. In this research, the non-isothermal crystallization kinetics of PA10T/10I-based TPAE were studied using DSC at various cooling rates and analyzed via Jeziorny’s equation and Mo’s method.

Figure 8 presents the DSC cooling curves of TPAEs and PA10T/10I at various cooling rates. The correlative crystalline parameters are summarized in Table 4. As shown in Figure 8, the melting peak slightly moves towards the low temperature region as the cooling rate increases. In Table 4, the value of enthalpy (Δ*H_c_*) decreases with cooling rate increase, meaning that high cooling rates hinder the crystallization of TPAEs. On the other hand, the value of *t*_1/2_ is relatively low at a high cooling rate, which can be explained by the fact that hard segment molecular chain does not have enough time to engage forming crystal structures in this condition. Otherwise, the crystallization rate at the same cooling condition can be derived from the rank of *t*_1/2_. The crystallization rates of resultant TPAEs and PA10T/10I followed order of TPAE-2000 > TPAE-1500 > PA10T/10I > TPAE-1000. The reasons might be that TPAE-2000 and TPAE-1500 have relatively long hard segment molecular chains which can result in strong intermolecular forces. In addition, low content of soft segment can improve the flexibility of TPAEs [38]. So, these two samples have higher crystallization rates than PA10T/10I. However, when the content of soft segment increases to 50 wt%, the regularity of molecular chains is disrupted, the molecular chain is difficult to arrange into crystals and the crystallization rate of TPAE-1000 is the slowest.

To study the non-isothermal crystallization kinetics of TPAEs, the values of relative crystallinity should be calculated based on the DSC curves. The relative crystallinity as a function of temperature is given in the equation below [40]:(1)$$X\left(T\right)\;=\;\frac{H_{T}}{{\Delta H}_{C}}\;=\;\frac{\int_{T_{0}}^{T}\left(dH_{c}/dT\right)dT}{\int_{T_{0}}^{\infty }\left(dH_{c}/dT\right)dT}\times \;100\%$$
where *H_T_* can be converted to the heat generated from temperature *T* to *T_0_*, and Δ*H_c_* is the heat generated in the whole crystallization period; *dH_c_* is the enthalpy in infinitesimal temperature range *dT*; *T_0_* and *T_∞_* are the initial and the end temperature of crystallization, respectively. The obtained results were plotted as curves and are shown in Figure 9.

The instantaneous crystallization temperature *T* and crystallization time *t* can be converted by applying the following equation [41]:(2)$$t\;=\;\frac{T_{0}-T}{R}$$
where *T* is the temperature at a crystallization time *t*; *T_0_* is the temperature when crystallization starts and *R* is the cooling rate.

A new equation of the relative crystallinity *X*(*t*) as a function of crystallization time *t* can be obtained by combing equations:(3)$$X\left(t\;\right)\;=\;\frac{\int_{t_{0}}^{t}\left(dH_{c}/dt\right)dt}{\int_{t_{0}}^{t_{\infty }}(dH_{c}/dt)dt}$$
where *t_0_* and *t_∞_* are the initial and the end times of crystallization, respectively; *dH_c_* is the enthalpy in infinitesimal time range *dt*. The relative crystallinity as a function of time is shown in Figure 10.

#### 3.5.1. Jeziorny Method

Initially, Avrami’s equation was used for analysis of the isothermal crystallization behavior. Mandelkern assumed that the crystallization temperature was constant and applied Avrami’s equation to analyze the primary stage of non-isothermal crystallization. The equation is presented as follows [42]:(4)$$1-X_{t}=\exp\left(-Z_{t}t^{n}\right)$$
where *Z_t_* is the rate constant in the non-isothermal crystallization process; *t* is the crystallization time; *n* is the Avrami exponent which depends on nucleation and dimension of crystallite growth. By taking the lg-lg linearity of Equation (4), Equation (5) can be obtained as follows [43]:(5)$$\;\mathrm{lg}\left[-\ln\left(1-\frac{X_{t}}{100}\right)\right]=\mathrm{lg}Z_{t}+n\mathrm{lg}t$$

The values for *n* and *Z_t_* can be attained from the slope and intercept of the plots of lg[−ln(1−*X_t_*/100)] vs. lg*t*, respectively. In order to make the equation more applicable to analyzing non-isothermal crystallization behavior, Jeziorny [44] considered the influence on the cooling rate *R* and came up with the final form of the rate parameter verifying the kinetics of non-isothermal crystallization, which was given as follows [45]:(6)$$\mathrm{lg}Z_{c}\;=\;\frac{\mathrm{lg}Z_{t}}{R}$$
where *Z_c_* is the revised crystallization rate constant. Another important parameter is the half-time of crystallization, *t*_1/2_, which is defined as the time to reach 50% of the relative crystallinity. The relation between the half-time and *Z_t_* can be expressed as follows [46]:(7)$$t_{1/2}\;=\;[\frac{\ln2}{Z_{t}}{]}^{1/n}$$

Figure 11 shows the curves of plots of lg[−ln(1−*X_t_*/100)] vs. lg*t* for the PA10T/10I and TPAEs. As shown in Figure 11, the curves show linearity at most of the stages; a deviation occurs only in the final phase of crystallization. Such deviations may correspond to spherulitic impingement and secondary crystallization. Therefore, the fitting area is selected in the relative crystallinity in the range of about 0–80%, and the corresponding results are listed in Table 5. From this table, it is worth noting that the Avrami exponent n of both PA10T/10I and TPAEs are in the range of 2~3, which means that these four samples present two- or three-dimensional crystallization growth throughout most of the crystallization period. Furthermore, the value of *Z_c_* increases with increasing cooling rate, which means the higher the cooling rate, the faster the crystallization rate. It can be explained that the free energy barrier of nucleation will decrease with increasing cooling rate, resulting in an increase in crystallization rate. Compared to the value of *Z_c_* at the same cooling rate, the order of these four samples follow TPAE-2000 > TPAE-1500 > PA10T/10I > TPAE-1000, agreeing well with the value of *t*_1/2_ in Table 5. Moreover, compared the *t*_1/2_ calculated from Avrami’s equation and the real *t*_1/2_ in Table 5, it can be found that these two values of TPAE are very close, but the values of PA10T/10I show a larger difference, meaning that the half-time of PA10T/10I calculated from Avrami’s method is not accurate.

#### 3.5.2. Mo’s Equation

In order to find a more appropriate model to depict the non-isothermal crystallization behavior, Mo and coworkers combined Avrami’s and Ozawa’s equations and suggested a new non-isothermal crystallization kinetic. It assumes that the relative crystallinity is associated with the cooling rate and crystallization time. The equation is given below [47,48]:(8)$$\ln Z_{t}\;+\;n\ln t\;=\;\ln K\left(T\right)-\;m\ln R$$
where *n* and *m* represent the Avrami and the Ozawa exponents, respectively. After variable substitution, the final new equation is as follows:(9)lgR = lgF(T)−αlgt
where *α* = *n/m*, the ratio between the Avrami and Ozawa exponents. The parameter *F*(*T*) = [*K*(*T*)/*Z_t_*]^1/*m*^ refers to the necessary value of cooling rate to achieve a determinate crystallinity at a unit of time for crystallization. Higher *F*(*T*) values mean that it needs more time to reach the end point.

According to the given degree of relative crystallinity, the plots of lg*R* as a function of lgt for PA10T/10I and TPAEs are presented in Figure 12. The intercept and slope of the fitting line are corresponding to ln*F*(*T*) and α, respectively, and their values are listed in Table 6. From Figure 12, the fitting line and plot are almost overlapped and the coefficient of determination *R*^2^ > 0.97, verifying the applicability of the combined method to this copolymer system. As shown in Table 6, it could be noted that with the increment in relative crystallinity, the value of *F*(*T*) increased. It is because the polymer chain has high segment motion ability at a low degree of crystallinity, while at a high value of *X_t_*, the chain mobility is hindered by the crystal structure which was formed before. In general, higher *F*(*T*) values mean that more time is required to achieve a definite degree of crystallinity at a certain cooling rate. That is, the higher the value of *F*(*T*), the slower the crystallization rate. Compared to the value of *F*(*T*) at the same relative crystallinity, the order of *F*(*T*) is TPAE-2000 > TPAE-1500 > PA10T/10I > TPAE-1000, which means that the order of crystallization rate follows with TPAE-2000 > TPAE-1500 > PA10T/10I > TPAE-1000. This result is consistent with the result of Jeziorny’s method and *t*_1/2_.

### 3.6. Activation Energy of Non-Isothermal Crystallization

Considering the influence of the various cooling rates in the non-isothermal crystallization process, Kissinger [49] reported that the activation energy Δ*E* could be determined as follows:(10)$$\frac{d[\ln\left(\frac{R}{T_{p}^{2}}\right)]}{d(\frac{1}{T_{p}})}=-\Delta E/R_{g}$$
where *T_p_* is the peak temperature, *R_g_* is the gas constant and Δ*E* is the activation energy. Equation (10) can be transferred into Equation (11).
(11)$$\ln\left(\frac{R}{T_{p}^{2}}\right)=\left(\frac{1}{T_{p}}\right)\left(\frac{-\Delta E}{R_{g}}\right)$$

The good linear relation plot of lg(*R*/$T_{p}^{2}$) vs. 1/*T_p_* is shown in Figure 13. The values of Δ*E* can directly be obtained from the slopes of the fitted lines, and the crystallization activation energies of TPAE-1000, TPAE-1500, TPAE-2000 and PA10T/10I correspond to −13.9, −22.3, −22.9 and −17.9 KJ/mol, respectively; the order of crystallization activation energies agrees with the results of the Avrami analysis. In other words, the introduction of PEG has a great influence on crystallization activation energies.

However, the Kissinger method has a disadvantage when estimating the crystallization activation energy of non-isothermal kinetics. As mentioned above, non-isothermal crystallization is a complex process that the Kissinger method calculates from a narrow temperature region and it only provides a single activation energy value to describe the whole crystallization process. A more accurate equation that can estimate activation energy at different degrees of crystallinity was proposed by Friedman [29]. The equation is shown as follows:(12)$$\ln{\left(\frac{dX}{dt}\right)}_{x}{=\;\ln\theta }_{t}\;=\;A-\frac{\Delta E_{x}}{{RT}_{x}}$$
where ln *dX_t_/dt* is the instantaneous crystallization rate at a given degree of crystallinity *X_t_*; Δ*E_x_* is the corresponding effective activation energy; *T_x_* is the temperature at relative crystallinity *X_t_*; *A* is a random pre-exponential coefficient. The effective activation energy can be obtained from line fitting of ln(*dX*/*dt*)*X* vs. 1/*T_X_* at different cooling rates.

Figure 14 shows the plots of ln(*dX_t_*/*dt*)*X_t_* as a function of 1/*T* at different relative crystallinities for PA10T/10I and TPAEs. The slope of linear fitting on the plot can be obtained to be Δ*E*/*R*, and the relationship between the obtained Δ*E* and the relative crystallization is shown in Figure 15. As can be seen in this figure, during the whole crystallization process, the value of Δ*E* is negative, indicating that the crystallization is a spontaneous process, and the higher the Δ*E*, the lower the crystallization ability of the polymer. In addition, the value of Δ*E* is decreased as the relative crystallinity increases, especially at regions with a high extent of relative crystallization. The reason for that is the hindrance of polymer crystals generated from molten molecular chain increases. Obviously, during the whole crystallization process, the order of Δ*E* of these four samples follows TPAE-2000 > TPAE-1500 > PA10T/10I > TPAE-1000, which is consistent with the results of the Jeziorny method and *t*_1/2_.

### 3.7. Crystal Morphology

Polarization microscope images of the TPAEs and PA10T/10I crystallized at 10 °C/min are shown in Figure 16. The crystals were generated from the crystallization of hard segment PA10T/10I. In Figure 16, it is difficult to find a typical spherulitic structure, which is due to the weak crystallization ability of PA10T/10I. From a comparison of these four pictures, the sizes of the crystals are different and follow the order of TPAE-2000 > TPAE-1500 > PA10T/10I > TPAE-1000. This trend is in agreement with the results of the non-isothermal crystallization kinetics. In addition, the polarizing micrographs under other cooling rates also show similar results (Appendix A).

## 4. Conclusions

Novel thermoplastic polyamide elastomers based on PA10T/10I were successfully synthesized via a facile one-pot method in this work. PA10T/10I was chosen to be the hard segment and PEG was chosen to be the soft segment. By changing the molecular weight of the hard segment, the mechanical properties of the TPAEs had significant differences. Specifically, as the molecular weight of the hard segment increased, the tensile strength of the TPAEs increased, while the elongation at break decreased. Controlled mechanical properties made this elastomer have more applications to meet various requirements. Thermal test results indicated that the TPAEs have double-melting behavior, and the melting temperatures are very close to that of the PA10T/10I hard segment. The thermo-stability of the TPAEs is better than most thermoplastic polyamide elastomers, and the high-temperature bending test results revealed that the TPAEs have better performance than traditional aliphatic polyamide elastomers. In addition, Avrami’s equation and Mo’s method were employed to investigate non-isothermal crystallization kinetics of the TPAEs and PA10T/10I. Together, both methods are suitable to analyze the kinetics of TPAEs, while Mo’s method has a wider range of applicability, especially for analyzing the kinetics of PA10T/10I. These two methods present similar results showing that as the molecular weight of hard segments increases, the half-times of non-isothermal crystallization and the crystallization rates follow the order of TPAE-2000 > TPAE-1500 > PA10T/10I > TPAE-1000. This trend was also proven by the inactivation energies which were calculated by Kissinger and Friedman methods. In future work, an attempt will be made to synthesize a TPAE with a higher 10T content or higher hard segment molecular weight. The effect of copolymerization proportion on crystallization kinetics is worthy of researching deeply.

## Data Availability

Data available on request due to restrictions eg privacy or ethical.

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
