# Peer review of "Synthesis, Characterization and Non-Isothermal Crystallization Kinetics of a New Family of Poly (Ether-Block-Amide)s Based on Nylon 10T/10I"

_polymers, 2020, doi:10.3390/polym13010072_

Round 1

Reviewer 1 Report

Review of article Manuscript ID: polymers-1041845

Title:  Synthesis, Characterization and Non-isothermal Crystallization Kinetics of a New Family of Poly (ether-block-amide)s Based on Nylon 10T/10I

Authors: Xin Tong , Zhao Wang , Mei-Ling Zhang , Xiao-Jun Wang , Gang Zhang , Sheng-Ru Long , Jie Yang

Presented manuscript is of the theoretical and practical significance.

The paper presents the synthesis and characterization of new thermoplastic elastomers with considerable mechanical properties.

Using the thermal analysis, studies regarding the non-isothermal crystallization analysis were made.

Regarding the content of the manuscript:

Abstract – Please reformulate the phrase: “The result of tensile test indicates these resultant elastomers have good mechanical properties with tensile strength ranging from 21.9~41.1 MPa”.

There are many grammatical errors: e.g. – page 2 –line 56 – “…PA9T and PA10T was found to….”

Page 2, line 87: “…in polymers can be concluded in two types: isothermal crystallization, and non-isothermal crystallization.” Please, reformulate the phrase.

Please, correct it.

Page 3- line 97 “…Jeziorny and Mo’s methods…” Please, give a reference.

The FT-IR results and the corresponding peaks must be correlated with literature.

Please reformulate the Introduction section by adding similar results from literature emphasizing the scientific significance of this study.

The reference list and the numbering in the manuscript should be checked and corrected according to the journal requirements.

I hence proposed "accept with revision".

Reviewer 2 Report

The manuscript, basically, describes the synthesis and characterization, by conventional procedures, of a family of elastomers by combining soft and hard blocks, in this way, controlling the mechanical and thermal properties of the resulting materials. Such a strategy, however, is well described in the literature, which makes the manuscript lacking in novelty. Moreover, a very similar work has been published very recently by the same authors (https://doi.org/10.1002/pi.6119). Also, an extensive english editing is needed in the whole manuscript. Based on the above, I do not support its publication in Polymers.

Reviewer 3 Report

Manuscript: Synthesis, Characterization and Non-isothermal Crystallization Kinetics of a New Family of Poly (ether-block-amide)s Based on Nylon 10T/10I

Manuscript presents good research work related to Poly (ether-block-amide)s Based on Nylon 10T/10I. It is recommended for publication after following minor changes.

  • Abstract should contain some quantitative information also.
  • English must be improved.
  • Novelty of the work be established.
  • All the results reported be compared in a tabular form to establish the superiority of the work.
  • Authors must need to incorporate some recent references in the introduction part of the manuscript related to the theme to make it more interesting for the readers.
  • Authors need to improve quality of figure 1.
  • Authors need to include future prospective of the work in the conclusion part of manuscript.

It will be good if authors can include some microscopy studies in the manuscript.

Round 2

Reviewer 2 Report

Based on the authors response, the manuscript can be published after the appropriate corrections suggested by all referees.